# Sustained Remission Off-Treatment (SROT) of TPO-RAs: The Burgos Ten-Step Eltrombopag Tapering Scheme

**DOI:** 10.3390/medicina59040659

**Published:** 2023-03-27

**Authors:** Tomás José González-López, Drew Provan

**Affiliations:** 1Hematology Department, Hospital Universitario de Burgos, 09006 Burgos, Spain; 2Academic Haematology Unit, Blizard Institute, Barts and The London School of Medicine and Dentistry, Queen Mary University of London, London E1 2BB, UK; a.b.provan@qmul.ac.uk

**Keywords:** tapering, sustained remission off-treatment (SROT), discontinuation, TPO-RAs, eltrombopag, romiplostim, avatrombopag

## Abstract

*Background and Objectives:* TPO-RAs (romiplostim/eltrombopag/avatrombopag) have broadly demonstrated high efficacy rates (59–88%), durable responses (up to three years) and a satisfactory safety profile in clinical trials. The effect of TPO-RAs is classically considered to be transient because platelet numbers usually dropped rapidly to baseline unless therapy was maintained. However, several groups have reported the possibility of successfully discontinuing TPO-RAs in some patients without further need for concomitant treatments. This concept is usually referred as sustained remission off-treatment (SROT). *Materials and Methods:* Unfortunately, we still lack predictors of the response to discontinuation even after the numerous biological, clinical and in vitro studies performed to study this phenomenon. The frequency of successful discontinuation is matter of controversy, although a percentage in the range of 25–40% may probably be considered a consensus. Here, we describe all major routine clinical practice studies and reviews that report the current position on this topic and compare them with our own results in Burgos. *Results:* We report our Burgos ten-step eltrombopag tapering scheme with which we have achieved an elevated percentage rate of success (70.3%) in discontinuing treatment. *Conclusions:* We hope this protocol may help successfully taper and discontinue TPO-RAs in daily clinical practice.

## 1. Introduction

Thrombopoietin receptor agonists (TPO-RAs; romiplostim/eltrombopag/avatrom bopag) have broadly demonstrated high efficacy rates (59–88%) and a satisfactory safety profile in clinical trials [1,2,3,4]. Durable responses after long-term continuous use (up to three years) of TPO-RAs were observed. Classical knowledge considered the effect of TPO-RAs to be transient because the platelet numbers usually dropped rapidly to the baseline unless therapy was maintained [5]. However, several groups have reported the possibility of successfully discontinuing TPO-RAs in some patients without further need for concomitant treatments. This concept is usually referred to as sustained remission off-treatment (SROT) [6,7,8,9].

Thus, short- and medium-term therapies with TPO-RA may reduce costs for our healthcare systems and, most importantly, may reduce the potential adverse effects associated with continuous treatment with TPO-RAs [8]. We face two major limitations when attempting to discontinue TPO-RAs in our daily routine practice: firstly, we lack predictors of a successful response to discontinuation despite numerous biological, clinical and even in vitro studies performed to study this phenomenon [7,8]. Secondly, the frequency of successful discontinuation is still a matter of debate, although a percentage in the range of 25–40% may probably be considered a consensus [8,9,10]. Here, we describe all major routine clinical practice studies and reviews reporting on the current position in relation to this topic [7,8,11]. Four important studies also deserve to be considered in detail: the Italian prospective experience regarding discontinuation in an early phase (newly diagnosed/chronic) of immune thrombocytopenia ITP [9], the French STOP-AGO trial [12], the International Taper trial [13] and the American Boston experience [14]. Three Delphi-based recommendations (Spanish, UK and Italian consensus) will also be reviewed [15,16,17]. 

## 2. Biological Basis for Discontinuation of TPO-RAs

TPO-RAs stimulate c-mpl (TPO-R, thrombopoietin receptor) and subsequently activate the Janus kinase (JAK)–signal transducer and activator of transcription (STAT) [JAK-STAT] downstream signaling [18]. Thus, they increase megakaryocytes in the bone marrow and peripheral blood platelet numbers soon afterwards. However, some preclinical and clinical studies suggest an immunomodulatory effect exerted by this type of drug with the obvious potential to affect the course of the disease (Figure 1) [19,20,21]. This effect is probably the main biological basis to support that SROT is potentially achievable when initiating a TPO-RA treatment at any ITP stage [9]. Here, we will explain the immunomodulatory basis for SROT.

Loss of self-tolerance is a feature of all autoimmune diseases, including ITP [22,23,24]. Immune dysregulation in ITP shows a shift towards a Th1 response (pro-inflammatory) with the elevation of IL-17-producing T helper (Th17) cells and increased interleukin (IL)-17 expression with a reduction in the B and T cell regulatory compartments (see Figure 1) [25,26]. The published data show that there is activation of CD4-positive T cells along with inhibition of T regulatory cells [27,28,29,30] and induction of megakaryocyte apoptosis [25,31].

Previous studies have shown that a sustained response off treatment is higher in the earlier phase of ITP compared to the later phase of ITP treated with TPO-RAs [6,32]. In patients who respond to TPO-RAs, there is an increase in transforming growth factor beta 1 (TGFb1) and the T regulatory cells with phagocyte inhibition [19,21]. In TPO-RA responders, there is an increase in the platelet mass, which results in an increase in the target platelet autoantigen. This may induce anergy of T cells with an overall reduction in the pro-inflammatory state [20,33]. In addition to the upregulation of TGFb1, there is also increased forkhead/winged helix transcriptional factor P3 (FoxP3) expression, which converts FoxP3 negative T regulatory cells to FoxP3 positive T regulatory cells. Soluble CD40 levels have been shown to be reduced [19,34]. 

The Fcg receptors on phagocytic cells also appear to play a key role in the pathophysiology of ITP and the induction of SROT, since there is a shift in the balance between inhibitory and activating Fcg receptors in patients responding to TPO-RAs. In ITP there has been shown to be an increased expression of FcgRI and IIa (activating receptors) with a reduction in FcgRIIb (inhibitory receptors) [35]. Previous studies have shown that there is a shift in the balance towards inhibitory FcgRIIb on monocytes with the associated reduction in phagocytic activity in patients who respond to treatment with TPO-RA [21]. Finally, administration of romiplostim in mice has shown that there is upregulation of inhibitory FcgRII expression with the down-regulation of FcgRI [21].

The precise role of the Fcg receptors is not known, but it would appear that there is a shift from the activated state towards an inhibitory state with an increase in the FcgRIIb receptors [35]. Clearly, more work needs to be done to elucidate the exact mechanism of the SROT and to confirm whether indeed this is restoration of immunological tolerance.

## 3. Tapering and SROT TPO-RAs: Major Publications, Clinical Approach to TPO-RA Discontinuation

Various guides and reviews support the discontinuation of TPO-RA [12,13,14,15,16,17,36] when there is lack of efficacy, there are unacceptable side effects (e.g., thrombosis, bone marrow fibrosis or hepatotoxicity) or successful remission after treatment is observed with the use of this type of treatment. It is worth noting here the new, interesting initiatives (iROM1 and iROM2 studies) taken by the Swiss group [37]. Regarding lack of efficacy, this is defined in the eltrombopag datasheet as no hematological response being observed after 4 weeks of treatment at a dose of 75 mg per day [38]. Romiplostim, however, should be discontinued earlier, i.e., when no platelet response is observed after 4 weeks of therapy with the maximal approved dose of 10 µg/kg/week [14]. The adverse event (AE) profiles observed among TPO-RAs may be different and this should be taken into account when the AE severity or its sustained character in time (even if the AE is mild) leads us to consider stopping TPO-RA treatment. In these cases, TPO-RA switching should be an alternate option [39].

SROT is, as mentioned earlier, an acronym for “sustained remission off-treatment”, which means successful discontinuation of TPO-RAs after tapering and discontinuation (Figure 1). One major topic is to establish which patients may be good candidates for successful TPO-RA discontinuation. Probably, those ITP patients, independently of the stage of the disease, who maintain stable platelet counts (50–100 × 10^9^/L) for a period of around 2–6 months may be suitable candidates for TPO-RA discontinuation [7,8,9,12,13]. Nevertheless, numerous groups have reported that there is not necessarily any minimum number of months of TPO-RA treatment before discontinuation [7,8], as SROT is possible even after only one month of treatment [7]. SROT was defined in an Italian prospective study as patients who did not relapse after a durable remission off-treatment, which they defined as 6 months. However, in Spain this was defined as 9 months [8], in the French STOPAGO trial as 24 months [12] and in the TAPER trial as 12 months [13]; so, there is still no consensus on how long the duration of discontinuation should be to consider a patient as successfully discontinued from TPO-RAs. Most studies published [7,8,12,13] attempt discontinuation of TPO-RA treatment only when the patient has previously attained a complete response (CR, platelets ≥ 100 × 10^9^/L). However, there are patients in whom tapering may be attempted where the platelet count is lower, for example 50–100 × 10^9^/L [9,16].

Although, often during TPO-RA treatment, we try to maintain a minimum TPO-RA efficacy dose (e.g., ≥20,000 platelets/µL), when attempting a TPO-RA SROT, there may be situations where advanced age, comorbidities, lifestyle, occupation or other medications might make tapering difficult [8]. For example, patients receiving antiplatelet agents or anticoagulant drugs would be at higher risk of bleeding during the tapering period. In these patients, it would be safer not to taper the TPO-RA and this should be discussed with the patient [37]. 

It is well known that the immune dysregulation associated with ITP is worse in studies of a late stage of the disease. This may be the rationale to expect better results with TPO-RAs in the early stages of ITP. Probably, and for the very same reason, it may also be easier to obtain SROT in newly diagnosed and persistent ITP than in chronic stages of the disease [9]. 

Here, we will summarise the major SROT studies in Table 1. Thus, Newland et al. reported their prospective discontinuation data regarding romiplostim use [6]. However, the French group with their eltrombopag or romiplostim TPO-RA discontinuation study (*n* = 20) was the first to report the possibility of SROT when TPO-RAs were used [7]. Among pioneering TPO-RA retrospective studies, the Spanish group has been the one with the highest number of eltrombopag discontinued patients reported to date (*n* = 80). In our eltrombopag case series, 26 of 49 evaluable patients (53%) showed sustained remission after 9 months [8]. Our Italian colleagues recently observed, in their prospective study, that 17/34 (50%) of adult newly diagnosed/persistent primary ITP patients who responded to a short course of eltrombopag were able to successfully discontinue this drug for at least six months soon afterwards [9]. Very importantly, Lucchini et al. obtained 50% SROT after an 8-month follow-up. Therefore, the results of our Italian colleagues confirm the data reported by the Spanish group, which are quite similar to ours [9]. In contrast, the Italian elderly trial has reported very different results (85.4% of SROT). Here Palandri et al., report SROT was attempted in 62/384 (16.5%) patients with 53/384 (13.8%) able to attain SROT for a median time of 1.3 years [11]. Higher SROT rates (56.2% and 41.9%, respectively) have been recently reported by the French [12] and International TAPER [13] studies, respectively. Among the limitations, it is important to note that all the studies had different follow-up periods (see Table 1); but, nonetheless, they all clearly show SROT can be achieved in a high percentage of patients who are candidates for discontinuation.

These contradictory results would need an accurate under-treatment-remission diagnosis in order to avoid ITP relapses when attempting to discontinue TPO-RA therapy after a platelet response is attained. Unfortunately, we are still not close to that diagnosis. Recently, the Boston group reported that the absence of direct glycoprotein-specific platelet autoantibodies in their 61 assays was highly sensitive and specific (87% and 91%, respectively) for clinical ITP remission on TPO-RA treatment (a negative test had a positive likelihood ratio of 9.5 for remission) [40,41,42]. Although they report that more positive antibodies predict a severer disease, they recognize that antibody serologic testing does not predict the treatment’s response to steroids, intravenous immunoglobulins (IVIg) or TPO-RAs [41]. Nevertheless, we suggest it may be worth considering autoantibody testing before attempting discontinuation because we hypothesize that our SROT rates may be much higher if we take this into account in our routine daily practice.

Numerous Delphi consensus guidelines suggest how to successfully discontinue TPO-RAs [15,16,17]. This current trending topic still has many questions with no clear answers. Nevertheless, most of these guidelines agree on some recommendations. These statements are summarized below. Thus, when a CR platelet count of (≥100 × 10^9^/L) is attained, the TPO-RA dose should be progressively tapered until potential discontinuation [15]. The duration and degree of a stable platelet response before attempting discontinuation is also a matter for discussion, i.e., 6–12 months under TPO-RA treatment with ≥75% platelet counts of >50 × 10^9^/L vs. ≥6 months of TPO-RA therapy with platelet counts of >100 × 10^9^/L without concomitant therapies as suggested in the UK Consensus [16] and Italian Consensus, respectively [17].

TPO analogs should never be stopped abruptly, and it is advisable to taper the drug over a variable period of time (from 2 to 3 [16] to 12 months, depending on the various publications), since the patient response to drug tapering is the most important factor in deciding the discontinuation schedule [37]. Resumption of TPO-RA treatment is advised by our UK colleagues if the ITP becomes symptomatic or the platelet count decreases [16]. This discontinuation failure is reported to be 29–35% in the UK ITP expert experience, which is lower than the rates reported by the Spanish and Italian groups [8,9]. The UK Delphi consensus also reports 86–106 days as the median time of failure in patients before reinitiating TPO-RAs [16]. No other consensus group has established any predicted timeframe for the retreatment of ITP after a discontinuation relapse (Table 1).

In order to allow the reader to have a better general idea of the factors that may anticipate retention of the response despite the discontinuation of therapy, we can summarise that, to date, the precise mechanism of the sustained response off-treatment remains unresolved; but, from the published data it would appear that patients who respond to TPO-RAs and achieve SROT shift from a pro-inflammatory to an anti-inflammatory state, with an increase in TGFb1 and restoration of the regulatory T and B-cell compartments, in addition to a down-regulation of pro-inflammatory cytokines [20,33,34,35]. Interestingly, the higher baseline levels of IL-4, IL-10, TNF-α, and osteopontin were considered negative factors predictive of a response in the ESTIT trial [9], while enrichment of a “TNFα signaling via NF-κB” signature with an overexpression of CD69 in the CD8+ T cells was observed in patients who did not respond to discontinuation [12].

Regarding romiplostim, it is advisable to reduce its dose by 1 μg/kg per week [37], and for eltrombopag we consider our own Burgos schedule. Thus, with eltrombopag we suggest initiating the discontinuation of eltrombopag (independently of the number of weeks under eltrombopag treatment) on the sole condition that the patient has a platelet count ≥80 × 10^9^/L in two weekly determinations. While platelet numbers are maintained at ≥30 × 10^9^/L during tapering, we advise patients to step down in our schedule table. If our patient has a platelet count of 20–30 × 10^9^/L, we will maintain the same step. If the platelet number drops under 20 × 10^9^/L, we will step up once. Platelet counts should be monitored twice weekly but the decision to step down, maintain dose or step up must be done at the beginning of the weekly cycle and only if the platelet numbers remain stable (and in the mentioned counts) in both prior week determinations. If a patient has to step up once, every next step down shall be done at least every other week instead of weekly, maintaining the same treatment step at least for two weeks. If there are two step ups during tapering, every next schedule change should be done at least monthly, maintaining the same treatment step at least for four weeks. Here, our Burgos ten-step eltrombopag tapering scheme consists of weekly repeated cycles as follow (Table 2). 

Our center has a registry of all patients diagnosed with ITP and who have received TPO-RA treatment since approval of this type of drug in 2011. Seventy-five patients were treated with eltrombopag in Burgos. Their median (interquartile range [IQR]) age was 70 (56–88) years; 53.3% were female. The median (IQR) time from diagnosis to eltrombopag initiation was 19 (1–376) months, the median (IQR) duration of exposure to eltrombopag was 8 (3–49) months and the median (IQR) eltrombopag dose was 54.3 (33.5–71.8) mg/day. 

Sixty-two patients (82.7%) achieved CR at least once: 53/63 were primary ITP patients and 9 were secondary ITP with 84.1% and 75% CR rates, respectively. In accordance with the current definitions, primary ITP patients were allocated to newly diagnosed (*n* = 7), persistent (*n* = 8) and chronic (*n* = 47) ITP groups. However, secondary ITP patients were allocated to ITP secondary to autoimmune diseases (*n* = 4), ITP secondary to neoplastic disorders (*n* = 4) and ITP secondary to viral infections (*n* = 4).

OF the 27 patients who attempted discontinuation in Burgos, we observed that tapering and discontinuation was achieved in 19 patients (70.4%) with no ITP relapses after a 12-month follow-up. Their median (interquartile range (IQR)) age was 60 (44–74) years with females making up 57.8% of the patients. The median (IQR) absolute increase in platelet counts from the baseline in the successfully discontinued patients was 73 × 10^9^/L (38–140). The major epidemiological characteristics were compared between patients who responded to discontinuation vs. non-responders; however, unfortunately, we did not observe any predictor of the response. 

Comparing age and sex from the population of responders to discontinuation vs. the whole population’s characteristics, we observed that these 19 successful patients are younger than the overall population (60 vs. 70 years) with a higher percentage of women (57.8% vs. 53.3%). 

As limitations, given that discontinuation was at the physicians’ discretion, some of our patients did not attempt discontinuation probably because our colleagues felt quite comfortable with the minimum efficient doses of eltrombopag. On the contrary, most of our patients did not try tapering and discontinuation because we observed that their eltrombopag requirements to maintain response were too high to advise tapering of eltrombopag.

## 4. Conclusions

The first point to stress is that successful TPO-RA discontinuation requires a degree of experience with TPO-RA use. When using this type of drug, most hematologists look for the minimum effective dose that is able to maintain safe platelet counts. Several groups (including ours) support this practice due to TPO analog tapering now being a frequent and successful practice. Thus, often tapering leads to discontinuation and we consider that when a TPO-RA is tapered with success, TPO-RA discontinuation should be tried soon afterwards.

Nevertheless, many questions have arisen regarding discontinuation. Firstly, what is the minimum platelet count to take into account before TPO-RA discontinuation in our clinical practice? Major discontinuation studies [7,8] have been performed only in CR patients. However, there is an interesting UK suggestion [6] to try discontinuation in patients with ≥50 × 10^9^/L. We propose ≥ 80 × 10^9^/L as the threshold for a discontinuation attempt. Secondly, what is the minimum duration that a patient should stay on a TPO-RA treatment to be considered for discontinuation? To date, there is no consensus regarding this topic. The French and Spanish publications [7,8] stated that no minimum treatment time was required for a patient to attain a durable response off-treatment. In contrast, other publications [6] do suggest a minimum duration of treatment in order to achieve SROT; however, we agree with Lucchini et al. [9] that this is an arbitrary point. We also agree with them that immune restoration is easier to restore in the early phase of ITP. With this in mind, a trend towards a higher SROT achieved early in the disease was hypothesised nd this has now been published [8]. Thirdly, predictors for discontinuation are lacking. All potential epidemiological issues have been investigated; so, we need to focus on biological studies to understand why some patients may benefit from discontinuation and others cannot. Fourthly, after many discontinuation studies, no consensus has yet been reached regarding the true percentage of patients who may able to stop TPO-RAs successfully. Note, a recent Italian prospective trial has confirmed our previous retrospective study: ± 50% of SROT [8,9]. Finally, what is the minimum time over which a patient must taper the dose in order to attain SROT? Since there is no evidence, no consensus agreement has been established to date. Our results with a discontinuation rate of 70.3% support our Burgos schedule as an efficient and successful scheme to be followed. In our case series, unfortunately, and probably because of our low discontinuation numbers, we did not find any predictor of the response. Nevertheless, these results are consistent with the publications from other groups. In our center, younger patients and females could achieve slightly higher response rates to discontinuation than other types of patients.

In conclusion, we must stress here that in this manuscript we are trying to highlight variations in clinical practice in terms of tapering regimens and discontinuation. Clearly, this needs to have a consensus view among ITP experts and be published in the major guidelines such as the American Society of Hematology Guidelines or The International Consensus. Hopefully with the updates underway for the guidelines, a tapering and discontinuation protocol will be added. This would be helpful for all clinicians treating ITP.

## Figures and Tables

**Figure 1 medicina-59-00659-f001:**
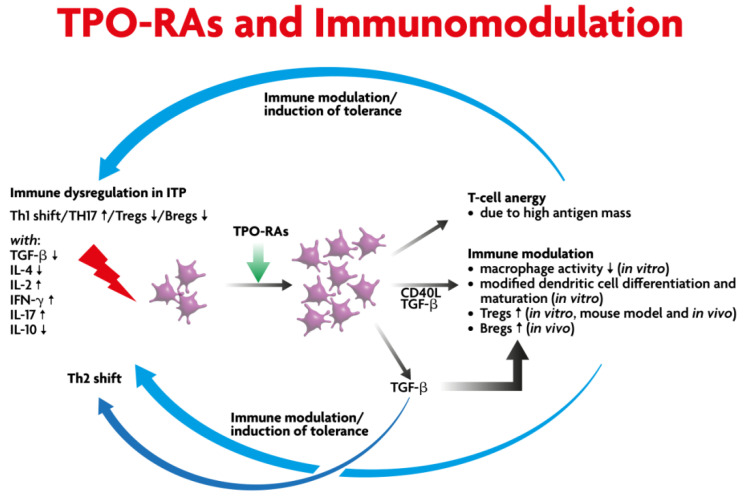
Scientific rationale for a durable sustained remission off-treatment (SROT) of TPO-RAs. (ITP: Immune thrombocytopenia; TH: T Helper; TGF: transforming growth factor; IL: interleukin; IFN: interferon).

**Table 1 medicina-59-00659-t001:** Summary of major SROT studies.

	Number of Patients Primarily Treated (*n*)	SROT Rate	Follow-Up Time (Months)	Type of TPO-RA Used	Median Number of Previous Treatments	Mean Time of TPO-RA Treatment before Tapering	Median Minimum Platelet Count at Tapering Initiation	Biological Investigation to Support Tapering (Yes/No)
Newland A, et al. (2016) [6]	98	21.3% (16/75)	6 months	Romiplostim	One	≤12 months	≥50 × 10^9^/L	No
Mahévas M, et al. (2014) [7]	54	50% (14/28)	13.5 months	Romiplostim and/or Eltrombopag	Six	5 years	≥300 × 10^9^/L	No
González-López TJ, et al. (2015) [8]	260	53.1% (26/49)	9 months	Eltrombopag	Four	46.5 months	≥100 × 10^9^/L	No
Lucchini E, et al. (2021) [9]	51	25% (13/51)	24 weeks	Eltrombopag	One	6 months	≥30 × 10^9^/L	Yes
Palandri, et al. (2021) [11]	390	16.5% (62/384)	9.6 years	Romiplostim or Eltrombopag	Not available	0.9 years	Not available	No
Mahevas M, et al. (2021) [12]	49	56.2% (27/48)	6 months	Romiplostim or Eltrombopag	Two	1.6 years	291 × 10^9^/L	Yes
Cooper N, et al. (2022) [13]	105	41.9% (44/105)	2 years	Eltrombopag	One	4 months	≥70 × 10^9^/L	No

**Table 2 medicina-59-00659-t002:** The Burgos ten-step eltrombopag tapering scheme.

First step (7 days-treatment)	50–50–25–50–50–25–50 (mg per day)
Second step (7 days-treatment)	50–25–25–50–25–25–50 (mg per day)
Third step (7 days-treatment)	25/24 hs (mg per day)
Fourth step (7 days-treatment)	25–25–0–25–25–0–25 (mg per day)
Fifth step (7 days-treatment)	25–0–0–25–0–0–25 (mg per day)
Sixth step (7 days-treatment)	25–0–0–0–25–0–0 (mg per day)
Seventh step (7 days-treatment)	25–0–0–0–0–12.5–0 (mg per day)
Eighth step (7 days-treatment)	25–0–0–0–0–0–0 (mg per day)
Ninth step (7 days-treatment)	12.5–0–0–0–0–0–0 (mg per day)
Tenth step (7 days-treatment)	No TPO-RA treatment

## Data Availability

Data supporting reported results can be found in the clinical database of the Haematology Department of the University Hospital of Burgos.

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
