# Peer review of "Sustained Remission Off-Treatment (SROT) of TPO-RAs: The Burgos Ten-Step Eltrombopag Tapering Scheme"

_medicina, 2023, doi:10.3390/medicina59040659_

Round 1

Author Response

Dear Medicina (Kaunas) journal staff:

Re: Sustained Remission off treatment of TPO-RAs: The Burgos Ten Step Eltrombopag Tapering Scheme

Thank you for sending our manuscript out for peer review. We are very grateful to the reviewers for their very useful comments and suggestions which will improve the paper considerably. We have incorporated these as discussed below. Obviously, their suggestions will greatly improve the manuscript. Thank you for that!

We very much hope that we have answered the reviewers’ queries sufficiently. If there are any additional queries, of course we would be very happy to answer those.

With best wishes

Yours sincerely

FIRST REFEREE

Comments to the Authors: This goal of the study was to describe all major routine clinical practice studies and reviews that report on discontinuing TPO-RAs in patients and also describe the Burgos ten step eltrombopag tapering scheme

  1. Line 159 I believe this is the first time you introduce CR so spell out on first use

Thank you for pointing this out. We will add the full term before the abbreviation on line 159.

  1. I don’t understand lines 158-161. I think the authors are saying the majority of studies are using a platelet response of 100K to determine the patient responded to the TPO but that theoretically even if the patient didn’t have a plt response over 100K the physician could still determine to take them off therapy

Thank you for this. We agree this is a little confusing; we will rephrase this to say

“Most studies published [7-8,12-13] attempt discontinuation of TPO-RA treatment only when the patient has previously attained a complete response (CR, platelets ≥100 x 109/L). But there are patients in whom tapering may be attempted where the platelet count is lower, for example 50-100 x 109/L [9,16]”

  1. The paragraph starting at line 162 is also hard to follow. I think the authors are trying to point out that all patients are not candidates for SROT especially if they have other factors for which they need a higher plt count. But I think that further confuses the issue of isn’t the whole goal of SROT that after receiving TPOs then you can come off and your plt count remains in safe levels?

Thank you for highlighting this. We agree this is confusing. We will rephrase thus:

Although, often during TPO-RA treatment, we try to maintain a minimum TPO-RA efficacy dose (e.g. ≥20,000 platelets/µl), when attempting a TPO-RA SROT there may be situations where advanced age, comorbidities, lifestyle, occupation, or other medications might make tapering difficult. For example, patients receiving antiplatelet agents or anticoagulant drugs would be at higher risk of bleeding during the tapering period. In these patients it would be safer not to taper the TPO-RA and this should be discussed with the patient.  [8]. [37]. [37].

  1. For the summary of studies who have looked at this Table 1 it would be important to include the follow-up time as this may be why there are such differences

As recommended, we have included now the follow-up data from different studies. Thank you very much for the suggestion.

  1. In the text it would seem that you should refer to table one when you are talking about the summary of studies and not retreatment of ITP after discontinuation relapse…unless I’m not understanding table 1.

Thank you for this suggestion. You are correct and Table 1 does summarise the major SROT studies. We have amended the text accordingly.

  1. Line 194 you introduce IVIg and need to spell out on first use

Thank you for highlighting this. We will correct this in the text.

  1. On limitations I think it’s important to note that all the studies had different follow-up periods

Thank you for this. We agree and we will add text to that effect.

  1. For the studies that you are summarizing did the patients all take Romiplostim or all Eltrombopag or a combination?

As you can see in Table 1, we have now included this data in the paper.

  1. At the end of this article I still have all the same questions I had at the beginning, are you all suggesting the next step is a more formal systematic review with a meta analysis or that experts come together and create a standardized protocol I’m not sure what the take home message is.

Thank you for this comment. We were trying to highlight variations in clinical practice in terms of tapering regimens and discontinuation. Clearly this needs to have a consensus view among ITP experts and published in the major guidelines such as the American Society of Hematology Guidelines or The International Consensus. Hopefully with the updates underway for the guidelines a tapering and discontinuation protocol will be added. This would be helpful for all clinicians treating ITP.

Reviewer 2 Report

Gonzalez-Lopez and Provan present a review of a number of studies and reviews that address discontinuation of TPO-RAs in ITP. They also present a homemade 10-step scheme for tapering and discontinuation of Eltrombopag mentioning that they have had a high degree of success discontinuing treatment with it.

I have some questions/comments as follow:

1.     The review focuses on investigating those original papers and reviews that have been focused on tapering and discontinuation of TPO-RAs.  Therefore, the introduction describing the biological basis of discontinuation with TPO-RAs can be reduced.

2.     An overview of all included manuscripts presented in an outline, evaluating the factors that may anticipate retention of response despite discontinuation of therapy, would allow the reader to have a better general idea. This overview would also allow better conclusions to be drawn.

3.     Table1 could be used to make this overview. The authors should describe in it: the type of TPO used, median number of previous treatments, the mean time of treatment with the TPO that caused the tapering, the median minimum platelet count for which start tapering in each publication, and if they have data for example at one year in which they documented the recurrence or need for treatment. They also could summarize whether any biological investigation was published to support tapering and discontinuation.

4.     Table 1. Add the references in the table of each of the works listed it.

5.     Table1. What do you mean by patients involved, I guess the authors mean treated. Please modify.

6.     The authors mention that they had a rate of 86.4% successful having attained SROT in 113 patients from a whole population of 131 ITP cases treated with eltrombopag. The authors have to show all data of these 131 patients and not only the conclusion.

7.     Table II. The Burgos Scheme. Tapering is already known as an option in ITP patients treated with TPO-RAs, however the authors show a proposal with specific doses and its reduction for Eltrombopag that they have used in 113 patients with success. The authors must show the data with scientific rigor to support this proposal.

Author Response

Dear Medicina (Kaunas) journal staff:

Re: Sustained Remission off treatment of TPO-RAs: The Burgos Ten Step Eltrombopag Tapering Scheme

Thank you for sending our manuscript out for peer review. We are very grateful to the reviewers for their very useful comments and suggestions which will improve the paper considerably. We have incorporated these as discussed below. Obviously, their suggestions will greatly improve the manuscript. Thank you for that!

We very much hope that we have answered the reviewers’ queries sufficiently. If there are any additional queries, of course we would be very happy to answer those.

With best wishes

Yours sincerely

SECOND REFEREE

Gonzalez-Lopez and Provan present a review of a number of studies and reviews that address discontinuation of TPO-RAS in ITP. They also present a homemade 10-step scheme for tapering and discontinuation of Eltrombopag mentioning that they have had a high degree of success discontinuing treatment with it.

I have some questions/comments as follow:

  1. The review focuses on investigating those original papers and reviews that have been focused on tapering and discontinuation of TPO-RAS. Therefore, the introduction describing the biological basis of discontinuation with TPO-RAs can be reduced.

Thank you for this suggestion. We have the description of the biological basis has been reduced.

  1. An overview of all included manuscripts presented in an outline, evaluating the factors that may anticipate retention of response despite discontinuation of therapy, would allow the reader to have a better general idea. This overview would also allow better conclusions to be drawn.

Thank you for this suggestion. We have included one paragraph with major biological findings reported to date.

  1. Table1 could be used to make this overview. The authors should describe in it: the type of TPO used, median number of previous treatments, the mean time of treatment with the TPO that caused the tapering, the median minimum platelet count for which start tapering in each publication, and if they have data for example at one year in which they documented the recurrence or need for treatment. They also could summarize whether any biological investigation was published to support tapering and discontinuation.

Thank you for this useful suggestion. We have included all those items in the table.

  1. Table 1. Add the references in the table of each of the works listed it.

Thank you. We have now added the references for works listed in the table.

  1. What do you mean by patients involved, I guess the authors mean treated. Please modify.

Yes, we were referring to patients treated and we have amended the manuscript accordingly.

  1. The authors mention that they had a rate of 86.4% successful having attained SROT in 113 patients from a whole population of 131 ITP cases treated with eltrombopag. The authors have to show all data of these 131 patients and not only the conclusion.

Because it was data supplied by colleagues, we do not have access to the full data set. Nevertheless, we agree it would have been useful. We present here our data on regards our 20 patients who attempted discontinuation.

  1. Table II. The Burgos Scheme. Tapering is already known as an option in ITP patients treated with TPO-RAS, however the authors show a proposal with specific doses and its reduction for Eltrombopag that they have used in 113 patients with success. The authors must show the data with scientific rigor to support this proposal.

Our response to this question is similar to our response to point 6. Because it was data supplied by colleagues, we do not have access to the full data set. We present here our data regarding this topic.

Round 2

Reviewer 2 Report

The authors have answered my questions, there is an improvement in the quality of the manuscript